# Ambiphilic boron in 1,4,2,5-diazadiborinine

Baolin Wang[1], Yongxin Li[2], Rakesh Ganguly[2], Hajime Hirao[1] & Rei Kinjo[1]

Boranes have long been known as the archetypal Lewis acids owing to an empty $p$-orbital on the boron centre. Meanwhile, Lewis basic tricoordinate boranes have been developed in recent years. Here we report the synthesis of an annulated 1,4,2,5-diazadiborinine derivative featuring boron atoms that exhibit both Lewis acidic and basic properties. Experimental and computational studies confirmed that two boron atoms in this molecule are spectroscopically equivalent. Nevertheless, this molecule cleaves C–O, B–H, Si–H and P–H bonds heterolytically, and readily undergoes [4+2] cycloaddition reaction with non-activated unsaturated bonds such as C=O, C=C, C≡C and C≡N bonds. The result, thus, indicates that the indistinguishable boron atoms in 1,4,2,5-diazadiborinine act as both nucleophilic and electrophilic centres, demonstrating ambiphilic nature.

[1] Division of Chemistry and Biological Chemistry, School of Physical and Mathematical Sciences, Nanyang Technological University, 21 Nanyang Link, Singapore 637371, Singapore. [2] NTU-SPMS-CBC Crystallography Facility, Nanyang Technological University, 21 Nanyang Link, Singapore 637371, Singapore. Correspondence and requests for materials should be addressed to H.H. (email: hirao@ntu.edu.sg) or R.K. (email: rkinjo@ntu.edu.sg).

As classical trivalent boranes inherently possess an unoccupied *p*-orbital on the boron centre, they have been widely utilized as electron-pair acceptors or Lewis acids in synthetic chemistry[1]. Apart from this line of research, isolable nucleophilic and low-valent boron species have also attracted great attention in recent years since the seminal work by Nozaki, Yamashita and co-workers[2–11]. Among them, by installing two carbene ligands, the Bertrand group and our group developed neutral tricoordinate organoboron species isoelectronic with amines[12–15]. Braunschweig and co-workers reported relevant species, including the first complex featuring two carbon monoxide ligands coordinating to a boron centre[16,17]. In these twofold base adducts of monovalent boron, electrons in the filled *p*-orbital of the boron centre are delocalized into the formally empty *p*-orbitals or the π*-orbitals of the ligands. Thus, the bonding situation therein is reminiscent of the σ donation and π back donation between a metal and a ligand in conventional transition metal complexes[18,19], and there seems to exist similarity between this type of boron and transition metals, although there is controversy regarding how to interpret such bonding in main group compounds[20–22].

Recently, it has been demonstrated that various nonmetallic systems based on *p*-block elements featuring both strong electron-donor and electron-acceptor sites or small highest occupied molecular orbital–lowest unoccupied molecular orbital (HOMO–LUMO) gap, may be utilized for small-molecule activation[23–28]. Depending on the number of sites that participate in the activation, these nonmetallic systems can be classified into two major types. The first type has a nucleophilic centre and an electrophilic centre as independent active sites. Representative examples include phosphine-borane-containing species, pioneered by Stephan *et al.*[29–31], in which the phosphorus centre acts as a Lewis base, whereas the boron centre acts as a Lewis acid. The second type possesses a single active site with an ambiphilic property. Bertrand and co-workers reported the first successful, facile splitting of dihydrogen and ammonia by (alkyl)(amino)carbenes bearing an ambiphilic divalent carbon centre with a lone pair of electrons and a vacant orbital[32]. In the boron series, only one boron derivative isoelectronic with singlet carbenes has been crystallographically characterized, which exhibits a considerable electrophilic property[33]. Construction of organoboron compounds containing ambiphilic elemental centres still remains extremely challenging[34]. To date, a system with two ambiphilic sites for small-molecule activation, having both of the features mentioned above, has never been described.

Very recently, we have synthesized a 1,3,2,5-diazadiborinine bearing two boron atoms that are spectroscopically inequivalent. We showed that these two boron centres in an aromatic ring cooperatively activate small molecules in which one of them behaves as a Lewis acid centre, whereas the other serves as a Lewis base centre[35,36]. The result demonstrated that incorporation of two boron atoms into an aromatic skeleton allows for an effective remote interaction between them through the π-system[37]. Herein, we report the synthesis of 1,4,2,5-diazadiborinine in which two equivalent boron atoms act as both nucleophilic and electrophilic centres; thus, the compound features ambiphilic nature.

## Results

### Synthesis and characterization of 2

Reaction of compound **1** with excess amounts of potassium graphite ($KC_8$) in benzene slowly proceeded under ambient condition, and after work-up, 1,4,2,5-diazadiborinine derivative **2** was obtained as an orange powder in 52% yield (Fig. 1a). The [11]B NMR spectrum of **2** shows a singlet at $\delta = 18.3$ p.p.m., which is shifted downfield with respect to that ($-1.9$ p.p.m.) of **1**. In the [1]H NMR spectrum, a peak for the

methyl groups on two nitrogen atoms was observed at $\delta = 2.90$ p.p.m., and two signals for the *CH* protons of the imidazole ring moieties appeared at $\delta = 6.07$ and 7.40 p.p.m. These data indicate the highly symmetric nature of the product **2**, which was further confirmed by an X-ray diffraction study (Fig. 1b). The annulated $B_2C_2N_2$ six-membered ring is essentially planar, and the boron atoms display trigonal–planar geometry with the N1–B1–C1′ bond angle of $112.50(15)°$ (the sum of the bond angles: $B1 = 359.75°$). Both the B1–C1′ (1.491(3) Å) and B1–N1 (1.458(2) Å) distances are significantly shorter than those (1.610(2) and 1.551(2) Å) in **1**. The N1–C1 distance of 1.403(2) Å is only slightly longer than the C1–N2 bond (1.397(2) Å). The C2–C3 bond (1.346(2) Å) lies in the range of typical double-bond distances of carbon–carbon bonds. These structural properties suggest the delocalization of electrons over the π-system including the six-membered $B_2C_2N_2$ ring, which can be represented by the average of the several canonical forms, including **2a–g** (Fig. 1c). In the electronic paramagnetic resonance spectrum of a benzene solution of **2**, no signal was detected at room temperature.

To investigate the electronic property of **2**, we carried out quantum chemical density functional theory calculations. Natural bond orbital analysis gave a Wiberg bond index value slightly < 1.0 for the B1–N1 bond (0.94). Meanwhile, Wiberg bond index values were > 1.0 for the B1–C1′ bond (1.19), the C1–N1 bond (1.13) and the C1–N2 bond (1.10), thus suggesting the partial double-bond character of these bonds. The HOMO of **2** exhibits a π-system over the six-membered $B_2C_2N_2$ ring with a node found between two CBN π-units and also has large amplitudes on the N atoms and the C=C moieties of the annulated imidazole rings (Fig. 1d). The LUMO is a π-type orbital that contains a mixture of C–B π-bonding interactions and the π-orbital of the phenyl ring on the B atoms. Natural population analysis shows that the two boron atoms possess the same charge of $+0.55$.

Figure 2 summarizes the nucleus-independent chemical-shift (NICS) values for **2**, a model compound **2′** and other related compounds for comparison. The NICS values for **2** and **2′** are less negative than that of annulated indole, but comparable to that of benzene, and more negative than those of other heterocycles, suggesting the considerable aromatic property of **2**. The resonance stabilization energy (RSE) value of parent 1,4,2,5-diazadiborinine **2″** estimated at the B3LYP/6-311 + G(d,p) level is $37.9$ kcal mol$^{-1}$ smaller than that of benzene (Supplementary Table 4)[38,39]. Since the RSE value of benzene is estimated to be about 34 kcal mol$^{-1}$, although the reported values vary depending on the method used[38,39], it can be inferred that the aromatic character of parent 1,4,2,5-diazadiborinine **2″** is significantly weak, in line with the less negative NICS values for **2″** than that of benzene (Fig. 2).

**Reactivity.** Compound **2** is thermally stable both in the solid state and in solutions at ambient temperature, but decomposes rapidly on exposure to air. To examine the reactivity, we first treated **2** with methyl trifluoromethanesulfonate (MeOTf). A stoichiometric amount of MeOTf was added to a benzene solution of **2** at ambient temperature. After removing the solvent under vacuum, **3** was obtained as a mixture of diastereomers (1:1) in 87% yield (Fig. 3). An X-ray diffraction study revealed the solid-state structure of one of the diastereomers. The methyl group is attached to one of the boron atoms, while an oxygen atom of the triflate forms a bond with the other boron atom (Fig. 4), suggesting that two spectroscopically indistinguishable boron atoms in **2** may act as both nucleophilic and electrophilic centres. Because two boron atoms interact with each other through the π-system, it is reasonable to predict that when one

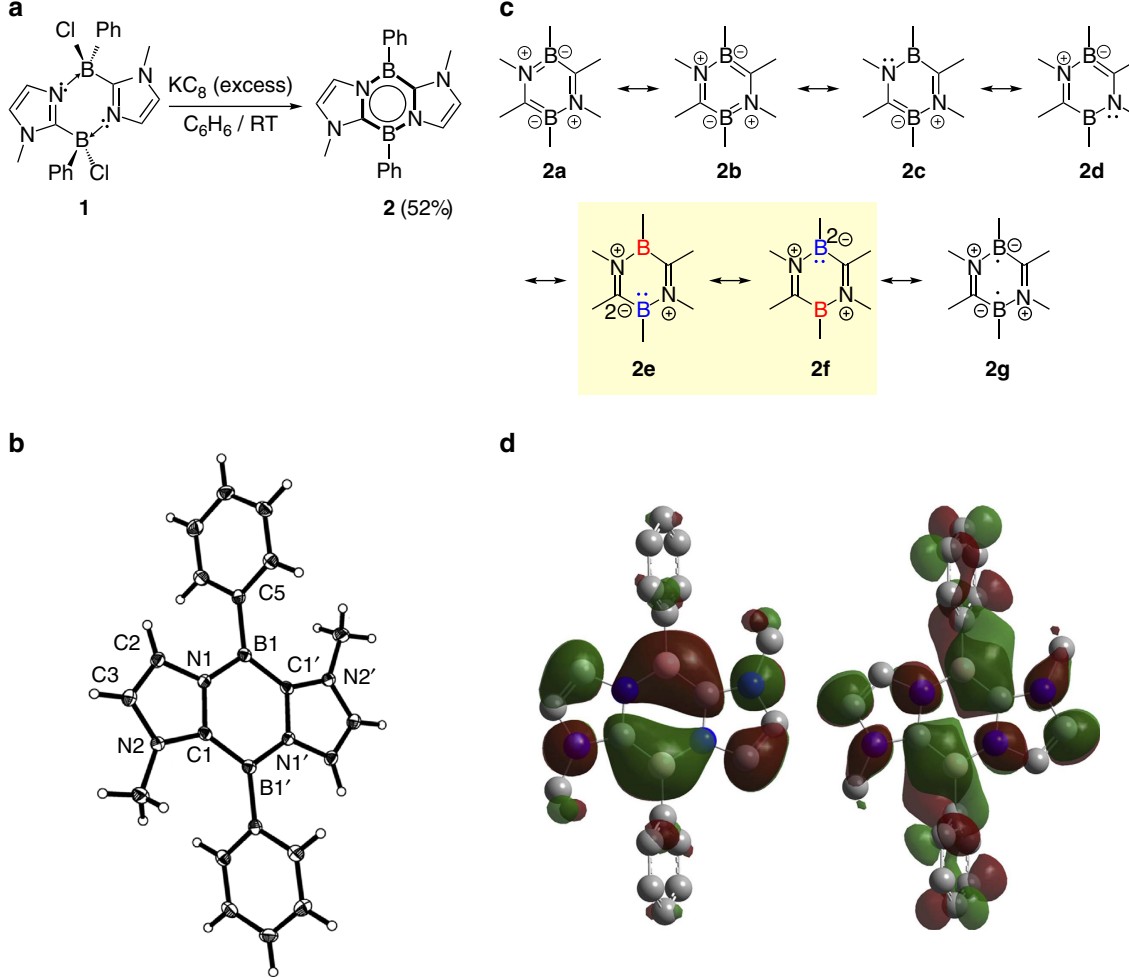

**Figure 1 | Characterization of annulated 1,4,2,5-diazadiborinine derivative 2.** (**a**) Preparation of **2** (Ph = phenyl). (**b**) Solid-state structure of **2** (thermal ellipsoids are set at the 50% probability level). (**c**) Schematic representations of selected canonical forms regarding the central $C_2B_2N_2$ ring of **2**. (**d**) Plots of the HOMO (left) and the LUMO (right) of **2**. Calculated at the B3LYP/6-311 + G(d,p) level of theory. Hydrogen atoms are omitted for clarity.

| | **2** | **2'** | | **2''** | | | |
|---|---|---|---|---|---|---|---|
| NICS(0) | −8.2 | −8.5 | −11.7 | −4.8 | −4.3 | −8.0 | −1.6 |
| NICS(1) | −8.0 | −8.8 | −11.9 | −6.9 | −6.5 | −10.2 | −2.7 |

**Figure 2 | Theoretical evaluation of aromaticity.** Calculated NICS(0) and NICS(1) values for **2**, **2'**, annulated indole derivative, parent 1,4,2,5-diazadiborinine **2''**, parent 1,3,2,5-diazadiborinine, benzene and borazine. Calculated at the B3LYP/6-311 + G(d,p) level of theory.

boron centre in **2** behaves as a Lewis base, the other boron centre takes on Lewis acidic character, as presented by the resonance forms **2e–f** (Fig. 1c). Having observed the ambiphilic nature of the boron atoms in **2**, we considered it likely that **2** would act as a frustrated Lewis pair (FLP)[30,31]. This hypothesis was borne out by further examination of the reactions between **2** and pinacolborane (HBpin) as well as arylsilane derivatives (PhSiH₃ and Ph₂SiH₂) because the reactions proceeded cleanly, and after work-up, adducts **4** and **5** were isolated in good yields (**4**, 83%; **5a**, 87%; **5b**, 85%). In contrast to the case of **3**, both products **4** and **5** were obtained as single diastereomers. **2** could readily activate the P–H bond of diarylphosphine

[($p$-FC₆H₄)₂PH] as well to afford product **6** in 85% yield. Products **3**, **4**, **5a** and **6** were fully characterized by standard spectroscopic methodologies and X-ray diffractometry (Fig. 4). In the solid-state structures of **4** and **5a**, the Bpin or Ph₂HSi group and the H atom on the B atom are attached on the same side of the six-membered B₂C₂N₂ ring. By contrast, the (FH₄C₆)₂P group and the H atom on the B atom in **6** point in opposite directions.

Next, we employed molecules involving non-activated unsaturated bonds as reactants. When $CO_2$ gas was introduced into a benzene solution of **2** at 1 bar, a white precipitate was instantaneously formed at room temperature. After work-up,

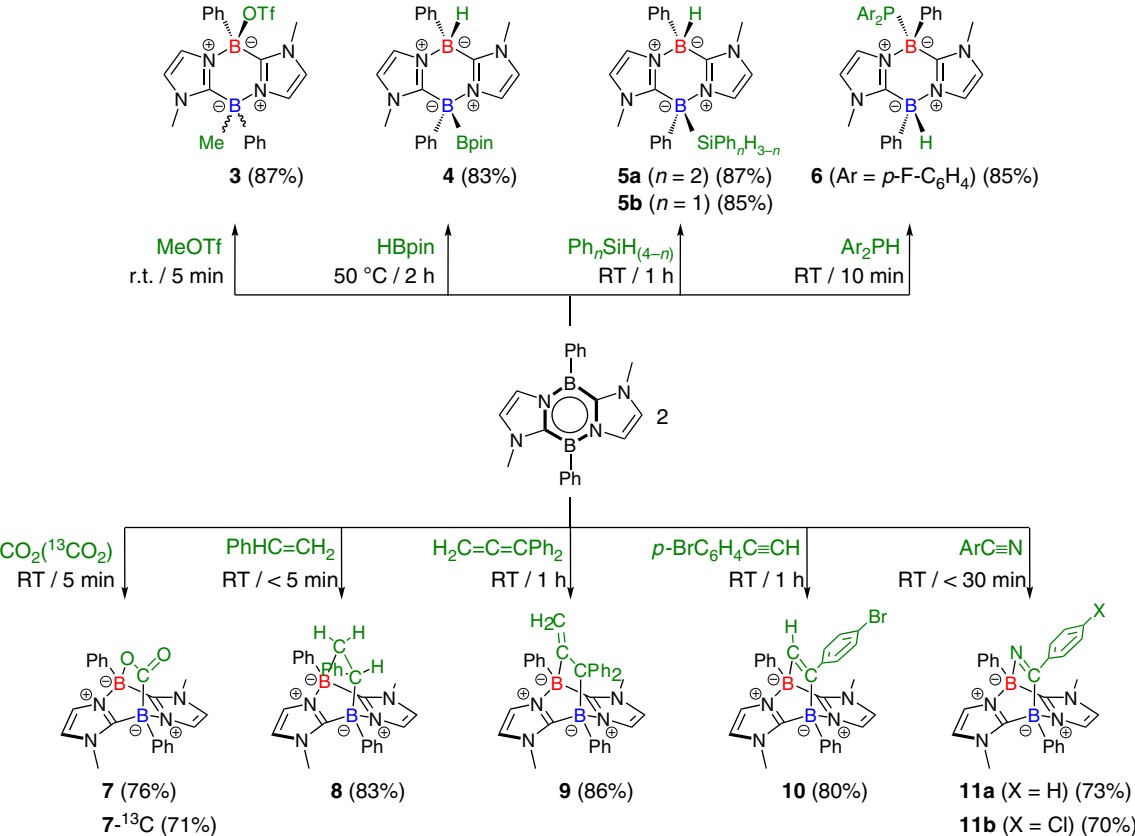

**Figure 3 | Reactivity of 2.** Reactions of **2** with MeOTf, HBpin, $Ph_nSiH_{(4-n)}$ ($n=1, 2$), $Ar_2PH$ (Ar = $p$-$FC_6H_5$), $CO_2$ ($^{13}CO_2$), $PhHC=CH_2$, $H_2C=C=CPh_2$, $p$-$BrC_6H_4C\equiv CH$, and $ArC\equiv N$ (Ar = Ph, $p$-$ClC_6H_4$).

compound **7** was isolated in 76% yield. We also performed a $^{13}C$-labelling study employing $^{13}CO_2$, which afforded **7-$^{13}C$** quantitatively. The $^{13}C$ NMR spectrum of **7-$^{13}C$** showed a broad singlet at 195.6 p.p.m. In the $^{11}B$ NMR spectrum, a set of new broad peaks was detected at $\delta = -1.0$ and $-10.8$ p.p.m. An X-ray diffraction study revealed the bicyclo[2.2.2] structure involving two boron atoms at the bridgehead, which was formed via [4 + 2] cycloaddition between **2** and one of the two C=O double bonds of $CO_2$. **2** was also able to activate styrene in a similar manner, to afford the corresponding bicyclo[2.2.2] derivative **8** stereoselectively as a single diastereomer in 83% yield. Analogously, the reaction of **2** with 1,1-diphenylpropa-1,2-diene ($H_2C=C=CPh_2$) proceeded smoothly, and clean formation of **9** through regioselective [4 + 2] cycloaddition was confirmed (86% yield), demonstrating a rare example of allene activation with a FLP[40]. Moreover, both alkyne and nitrile derivatives ($p$-$BrC_6H_4C\equiv CH$, $PhC\equiv N$ and $p$-$ClC_6H_4C\equiv N$) instantly reacted with **2**, to provide **10** (80% yield), **11a** (73% yield) and **11b** (70% yield), respectively. All of the cycloadducts **8–10** and **11b** were thoroughly characterized by various spectrometric methods and X-ray crystallography (Fig. 4).

**Proposed mechanism based on DFT calculations.** To gain insight into the reaction mechanism, pathways for the diastereo-selective formation of **4**, **5a**, **8** and **9** were explored computationally using Gaussian 09 at the B3LYP-D3(BJ) (SCRF)/6-311 + G(d,p)//B3LYP/6-311 + G (d,p) level of theory (Fig. 5)[41–53]. The default SCRF method was used to describe the solvent effect of benzene. For the formation of **4** and **5a**, both concerted and stepwise pathways were examined. However, only concerted pathways could be determined (Fig. 5a,b), suggesting

that these reactions proceed in a concerted mechanism. This explains why the H atom on the B atom and the Bpin group in **4** or the $Ph_2HSi$ group in **5** are attached on the same side of the six-membered $B_2C_2N_2$ ring. The relatively high free energy barrier (22.0 kcal mol$^{-1}$) for the formation of **4** is consistent with the fact that elevated temperature (50 °C) was required to accelerate the reaction. We compared the energies of **4** and **5** with their respective diastereomers **4★** and **5★** that possess the H atom on the B atom and Bpin group (**4★**) or $Ph_2HSi$ group (**5★**) on the opposite side of the $B_2C_2N_2$ ring. Both **4** and **5** are only slightly more stable than **4★** ($+0.4$ kcal mol$^{-1}$) and **5★** ($+1.8$ kcal mol$^{-1}$), respectively, indicating that the reactions may not be thermodynamically controlled.

For the reaction of **2** with styrene, we could determine two plausible concerted pathways, which revealed that the formation of **8** is favoured both thermodynamically and kinetically (Fig. 5c). Thus, product **8** is 0.6 kcal mol$^{-1}$ more stable than its diastereomer **8★**, and the activation barrier for the formation of **8** is 1.9 kcal mol$^{-1}$ lower than that of **8★**. Importantly, when the dispersion effect was not included, the difference in barrier height between the two pathways was only 0.4 kcal mol$^{-1}$ (Supplementary Table 6), indicating the important role played by attractive dispersion interactions, especially between the phenyl group of styrene and the methyl group of **2**, in determining the diastereoselectivity. As shown in Fig. 1d, both the HOMO and LUMO of **2** have significant amplitude over the B–C moieties rather than over the B–N moieties of the $B_2C_2N_2$ six-membered ring. At the transition state, hence, to maximize the interaction between the frontier orbitals of **2** and the $\pi$ or $\pi^*$ orbital of styrene, the C=C bond of styrene does not lie completely parallel to the line connecting the two B atoms of **2**, but the two carbon atoms are directed slightly towards the

**Figure 4 | Structural characterization of products.** Solid-state structures of **3**, **4**, **5a**, **6**–**10** and **11b**.

midpoints of the two B–C bonds of **2**. In such pathways, directing the phenyl group of styrene towards the B atom of **2** would cause significant steric repulsion between the Ph rings of styrene and **2**. The transition state for the favoured pathway looks less sterically encumbered, which could also contribute to the diastereoselectivity of the cycloaddition. Figure 5d shows the pathways obtained for the concerted cycloaddition between **2** and $H_2C=C=CPh_2$. The lower-energy pathway for the formation of **9** has an energy barrier of $15.1 \, kcal \, mol^{-1}$, which is $1.0 \, kcal \, mol^{-1}$ lower than that for the stereoisomer **9***. Interestingly, the pathway to **9*** has a lower barrier when dispersion is not taken into account, but the relative energy of the transition states is inverted on inclusion of the dispersion effect (Supplementary Table 6). This result again highlights the important role of dispersion in determining the regioselectivity. Compound **9** is $7.9 \, kcal \, mol^{-1}$ less stable than **9***, and thus formation of **9** is kinetically preferred.

The formation of compound **3** as a mixture of two diastereomers suggests that a stepwise reaction mechanism may

be involved, in addition to a concerted pathway (Fig. 6). However, our calculations could determine only a concerted pathway that gives **3B** (Supplementary Fig. 59). This might be attributed to the limited accuracy of density functional theory (DFT). We infer that the initial step would be methylation of **2** by MeOTf to afford an ionic intermediate **INT-3** (Fig. 6a). In the second step, triflate ($^-$OTf) could attack the tricoordinate boron centre of **INT-3** from either the same side or the opposite side of the $B_2C_2N_2$ ring, to form a B–O bond. The former attack would afford **3A**, whereas the latter would give **3B**. Compound **3A** is estimated to be $0.2 \, kcal \, mol^{-1}$ more stable than **3B**. Similarly, compound **6** may be formed through a stepwise mechanism because the $(FH_4C_6)_2P$ group and the H atom on the B atom in **6** point in opposite directions (Fig. 6b). However, DFT calculations could determine only a concerted pathway to afford **6*** (Supplementary Fig. 59), and thus the reaction mechanism for the formation of **6** remains unclear, although again, a stepwise mechanism might operate here. The possible stepwise pathways for the formation of **3** and **6** are in contrast to the concerted one in the formation of **4** and **5**.

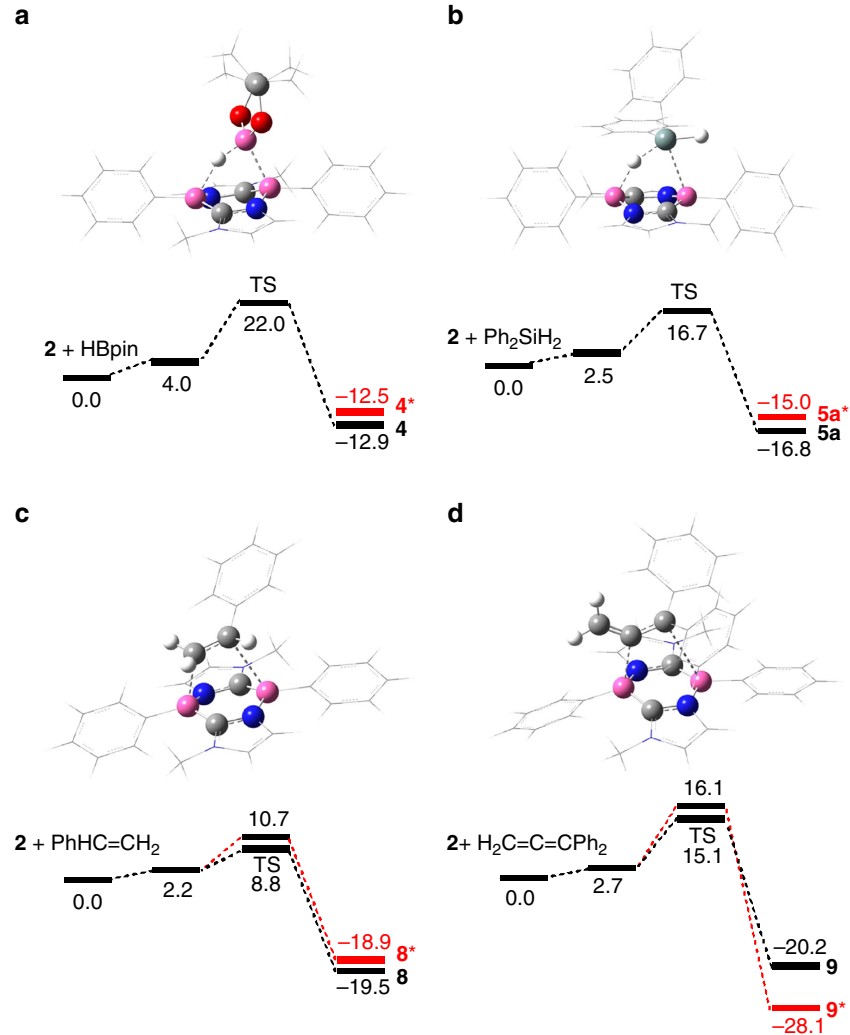

**Figure 5 | DFT-calculated free energy profiles of plausible concerted mechanism.** Energy profiles of possible mechanism for the stereo- and region-selective formation of **4**, **5a**, **8** and **9** from **2** with relative Gibbs free energies in kcal mol$^{-1}$ obtained at the B3LYP-D3(BJ)(SCRF)/6-311 + G(d,p)//B3LYP/6-311 + G(d,p) level, and dispersion force, all the compound numbers are in conjunction with Fig. 3). (**a**) Pathway for the formation of **4**. (**b**) Pathway for the formation of **5a**. (**c**) Pathways for the formation of **8** and the other diastereomer **8★**. (**d**) Pathways for the formation of **9** and the other diastereomer **9★**.

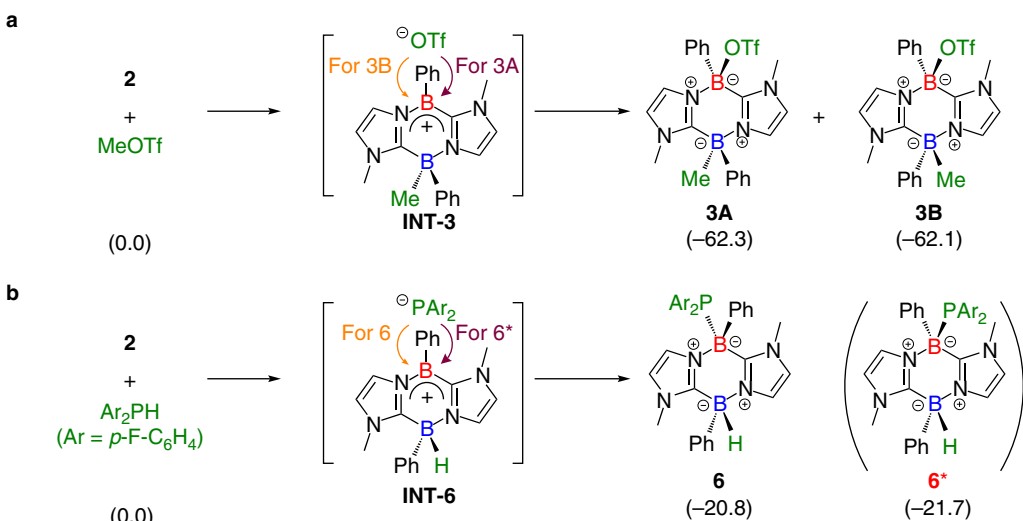

**Figure 6 | Proposed stepwise mechanism for the formation of 3 and 6.** Relative Gibbs free energies in kcal mol$^{-1}$ (estimated by optimization obtained at the B3LYP-D3(BJ)(SCRF)/6-311 + G(d,p)//B3LYP/6-311 + G(d,p) level. (**a**) Stepwise mechanism for the formation of **3A** and **3B** via **INT-3**. (**b**) Stepwise mechanism for the formation of **6** via **INT-6**.

Because the H atoms in H-Bpin and $Ph_2SiH_2$ are hydridic, the formation of the corresponding ionic intermediates such as $[2\text{-}H]^- [E]^+$ (E = Bpin or $SiPh_2H$) would be disfavoured due to the instability of the boryl or silyl cation fragment, which could be, at least in part, the origin of different stereo-selectivity observed in the formation of these products **4**, **5** and **6**. Isolation of ionic species as **INT-3** and **INT-6** is a subject of our ongoing study.

## Discussion

Stephan and co-workers reported that an aromatic triphosphabenzene featuring a $C_3P_3$ six-membered ring activates a hydrogen molecule under relatively mild conditions to give a [3.1.0]bicylo reduction product[54]. It has also been computationally confirmed that the initial transition state involves the deformation of the flexible $C_3P_3$ ring to a boat configuration. Similar boat character is predicted in the transition structure for the uncatalysed 1,4-hydrogenation of benzene[55]. Likewise, at the transition states for the formation of **4**, **5a**, **8** and **9** (Fig. 5), we observed distortion of the $B_2C_2N_2$ six-membered ring moiety in **2**, which exhibits some boat-like character. The less significant deformation is probably due to the lack of innate flexibility of the $B_2C_2N_2$ ring skeleton. It is inferred that such deformation may polarize frontier orbitals of **2** and enhance zwitterionic nature at two boron atoms, which would allow them to play cooperatively as a nucleophilic and an electrophilic centres.

Previously, we reported reversible [4 + 2] cycloaddition of 1,3,2,5-diazadiborinine with $CO_2$ as well as alkenes including ethylene[35,36]. A significant relevant example showed that 1-bora-4-tellurocyclohexa-2,5-diene undergoes subsequent [4 + 2] cycloaddition/alkyne-elimination via a Te/B FLP-type mechanism, reported by Stephan et al.[56]. These results prompted us to examine the thermal stability of the [4 + 2] cycloadducts **7** and **8**. A $C_6D_6$ solution of **7** or **8** in a degassed J-young NMR tube was heated and monitored by NMR spectroscopy. However, even at 150 °C, neither retro-[4 + 2] cycloaddition nor a pronounced decomposition was observed, demonstrating considerable stability of these cycloadducts. This is in good agreement with the greater computed thermodynamic stability of **8**.

Collectively, we have developed a synthetic protocol for aromatic 1,4,2,5-diazadiborinine **2**. According to the NICS values of **2**, **2′**, **2″** and benzene as well as the gap of the RSE values between **2″** and benzene, the annulation of the $B_2C_2N_2$ ring in 1,4,2,5-diazadiborinine may increase the aromatic nature of **2′**, which is comparable to that of benzene. Nevertheless, **2** exhibits reactivity that is peculiar and much higher than that of benzene. Experimental and computational studies on the reactivity demonstrate a manifesting dual character of the boron atoms in **2**, which is capable of activating a series of small molecules under mild reaction conditions. Such chemical behaviour has never been observed for the previously reported $B_2C_2N_2$ heterocycles[57–59]. As the substituents on the boron and nitrogen atoms in 1,4,2,5-diazadiborinine can be readily modified, the steric and electronic properties as well as the reactivity could be finely tunable. The isolation of this molecule provides a new strategy for the development of myriad B/N-containing $\pi$-systems, which involve two equivalent boron centres displaying ambiphilic nature. This approach could be of particular interest to the future design of various main group compounds featuring a FLP-type property.

## Methods

**Materials.** For details of spectroscopic analyses of compounds in this manuscript, see Supplementary Figs 1–55. For details of X-ray analysis, see Supplementary Tables 1–3, Supplementary Methods and Supplementary Data 1–11. For details of density functional theory calculations, see Supplementary Figs 56–59, Supplementary Tables 4–8 and Supplementary Methods.

**General synthetic procedures.** All reactions were performed under an atmosphere of argon or nitrogen using standard Schlenk or dry box techniques; solvents were dried over Na metal, K metal or $CaH_2$. Reagents were of analytical grade, obtained from commercial suppliers and used without further purification. $^1H$, $^{11}B$, $^{13}C$ and $^{19}F$ NMR spectra were obtained with a Bruker AVIII 400 MHz BBFO1 spectrometer at 298 K unless otherwise stated. NMR multiplicities are abbreviated as follows: s = singlet, d = doublet, t = triplet, m = multiplet and br = broad signal. Coupling constants $J$ are given in Hz. Most of signals for the quaternary carbon atoms bonding to boron atom could not be detected, presumably due to coupling with the B atom. Electrospray ionization (ESI) mass spectra were obtained at the Mass Spectrometry Laboratory at the Division of Chemistry and Biological Chemistry, Nanyang Technological University. Melting points were measured with an OpticMelt Stanford Research System. Infrared spectra were measured with the Bruker Alpha-FT-IR Spectrometer with an ECO-ATR module. Continuous wave X-band electron paramagnetic resonance (EPR) spectrum was checked using a Bruker ELEXSYS E500 EPR spectrometer.

**Synthesis of 1.** To a tetrahydrofuran (THF; 20 ml) solution of 1-methyl-1H-imidazole (0.55 g, 6.7 mmol), n-butyl lithium (1.6 M in hexane; 4.6 ml, 7.4 mmol) was added dropwise at − 40 °C. The reaction mixture was allowed to warm to room temperature over 60 min, and then transferred to a THF (40 ml) solution of dimethyl phenylboronate (1.0 g, 6.7 mmol) at − 78 °C. The mixture was warmed to room temperature and stirred overnight. Then, trimethylsilane chloride (1.09 g, 10 mmol) was added to the mixture at − 78 °C, and the solution was slowly warmed to room temperature and stirred for 5 h. After removal of all the volatiles, the residual solid was re-dissolved in dichloromethane (DCM; 40 ml). To the solution, boron trichloride (1 M in hexane; 6.7 ml, 6.7 mmol) was added at 0 °C, and the reaction mixture was warmed to room temperature, and stirred for 1 h. After filtration, all volatiles were removed under vacuum, and then the residue was washed with hexane and dried under vacuum to afford compound **1** as a pale yellow powder (1.02 g, 75%). Melting point: 235 °C (dec.). $^1H$ NMR (400 MHz, $C_6D_6$): δ = 7.95 (m, 4H, Ar-H), 7.30 (t, 4H, $J = 7.4$ Hz, Ar-H), 7.18 (t, 2H, $J = 7.4$ Hz, Ar-H), 6.76 (d, 2H, $J = 1.8$ Hz, CH), 5.52 (d, 2H, $J = 1.8$ Hz, CH) and 2.84 (s, 6H, N-$CH_3$); $^{13}C\{^1H\}$ NMR (100.56 MHz, $C_6D_6$): δ = 133.0 (Ar-CH), 128.3 (Ar-CH), 127.5 (Ar-CH), 123.0 (CH), 122.2 (CH) and 34.8 (N-$CH_3$); and $^{11}B\{^1H\}$ NMR (128.3 MHz, $C_6D_6$): δ = − 1.9 (s). High resolution mass spectrometry (ESI): $m/z$ calculated for $C_{20}H_{21}B_2Cl_2N_4$: 409.1329 $[(M+H)]^+$; found: 409.1338.

**Synthesis of 2.** Potassium graphite (3.76 g, 27.8 mmol) was added into a benzene (80 ml) solution of **1** (0.71 g, 1.74 mmol) at room temperature, and the reaction mixture was stirred for 5 days. After removal of graphite and inorganic salt by filtration, the solvent was removed under vacuum to afford **2** as an orange solid (305 mg, 52%). **2** decomposes at 155 °C without melting. $^1H$ NMR (400 MHz, $C_6D_6$): δ = 7.79 (m, 4H, Ar-H), 7.40 (m, 6H, Ar-H × 4 and CH × 2), 7.32(t, 2H, $J = 7.4$ Hz, Ar-H), 6.07 (d, 2H, $J = 1.9$ Hz, CH) and 2.90 (s, 6H, N-$CH_3$); $^{13}C\{^1H\}$ NMR (100.56 MHz, $C_6D_6$): δ = 135.4 (Ar-CH), 127.9 (Ar-CH), 126.9 (Ar-CH), 124.1 (CH), 113.8 (CH) and 36.1 (N-$CH_3$); and $^{11}B\{^1H\}$ NMR (128.3 MHz, $C_6D_6$): δ = 18.3 (s). Ultraviolet–visible (ε, in hexane): λ = 496 nm (7,600), 468 nm (4,280), 354 nm (2,140) and 273 nm (3,410). HRMS (ESI): $m/z$ calculated for $C_{20}H_{21}B_2N_4$: 339.1952 $[(M+H)]^+$; found: 339.1962.

**Synthesis of 3.** MeOTf (0.060 ml, 0.53 mmol) was added into a benzene (2.0 ml) solution of **2** (180 mg, 0.53 mmol) at room temperature, and the reaction mixture was stirred for 5 min. After removal of the solvent, the solid residue was washed with hexane and dried under vacuum to afford a yellow solid of **3** as a 1:1 mixture of diastereomers (216 mg, 87%). Recrystallization of **3** from a mixture of dichloromethane and hexane solution afforded single crystals of one of the diastereomers, which was confirmed by an X-ray diffraction analysis. When the single crystals were re-dissolved in $C_6D_6$ and checked by NMR spectroscopy, two diastereomers were observed in 1:1 ratio, suggesting that these two diastereomers are in equilibrium in solution at room temperature (For the estimated energy difference between two diastereomers, see Supplementary Figure 57.) *Mixture of two diastereomers of **3**: $^1H$ NMR (400 MHz, $C_6D_6$): δ = 7.80 (d, 2H, $J = 6.8$ Hz, Ar-H), 7.64 (d, 2H, $J = 6.8$ Hz, Ar-H), 7.54 (d, 2H, $J = 6.8$ Hz, Ar-H), 7.48 (d, 2H, $J = 6.8$ Hz, Ar-H), 7.37 (d, 2H, $J = 7.5$ Hz, Ar-H), 7.31–7.18 (m, 10H, Ar-H), 6.73 (d, 1H, $J = 1.6$ Hz, CH), 6.64 (d, 1H, $J = 1.6$ Hz, CH), 6.50 (d, 1H, $J = 1.6$ Hz, CH), 6.42 (d, 1H, $J = 1.6$ Hz, CH), 5.75 (dd, 2H, $J = 3.3, 1.6$ Hz, CH), 5.68 (t, 2H, $J = 1.6$ Hz, CH), 3.00 (s, 3H, N-$CH_3$), 2.90 (s, 3H, N-$CH_3$), 2.59 (s, 3H, N-$CH_3$), 2.57 (s, 3H, N-$CH_3$), 0.80 (s, 3H, B-$CH_3$) and 0.61 (s, 3H, B-$CH_3$); $^{13}C\{^1H\}$ NMR (100.56 MHz, $C_6D_6$): δ = 134.0 (Ar-CH), 133.2 (Ar-CH), 131.7 (Ar-CH), 131.4 (Ar-CH), 128.6 (Ar-CH), 128.5 (Ar-CH), 128.3 (Ar-CH), 128.2 (Ar-CH), 128.0 (Ar-CH), 127.9 (Ar-CH), 126.8 (Ar-CH), 126.5 (Ar-CH), 123.0 (CH), 122.9 (CH), 122.8 (CH), 122.7 (CH), 122.5 (CH), 122.4 (CH), 121.4 (CH), 121.0 (CH), 34.7 (N-$CH_3$), 34.5 (N-$CH_3$), 34.4 (N-$CH_3$) and 34.3 (N-$CH_3$); $^{11}B\{^1H\}$ NMR (128.3 MHz, $C_6D_6$): δ = 1.0 (br) and –7.7 (br); and $^{19}F\{^1H\}$ NMR (376 MHz,

$C_6D_6$): $\delta = -77.7$ and $-77.9$. HRMS (ESI): $m/z$ calculated for $C_{22}H_{24}B_2F_3N_4O_3S$: 503.1707 $[(M+H)]^+$; found: 503.1715.

**Synthesis of 4.** Pinacolborane (0.087 ml, 0.59 mmol) was added to a benzene (2.0 ml) solution of **2** (200 mg, 0.59 mmol) at room temperature, and the reaction mixture was stirred for 2 h at 50 °C. After removal of the solvent, the solid residue was washed with hexane and dried under vacuum to afford **4** as a yellow powder (229 mg, 83%). Melting point: 159 °C (dec). $^1$H NMR (400 MHz, $C_6D_6$): $\delta = 7.86$ (d, 2H, $J = 7.2$ Hz, Ar-$H$), 7.70 (d, 2H, $J = 7.2$ Hz, Ar-$H$), 7.39 (t, 2H, $J = 7.2$ Hz, Ar-$H$), 7.33 (t, 2H, $J = 7.2$ Hz, Ar-$H$), 7.26–7.20 (m, 2H, Ar-$H$), 7.12 (s, 1H, C$H$), 6.75 (s, 1H, C$H$), 5.92 (s, 1H, C$H$), 5.87 (s, 1H, C$H$), 3.20 (s, 3H, N-C$H_3$), 2.68 (s, 3H, N-C$H_3$) and 1.09 (s, 12H, C$H_3$); $^{13}$C{$^1$H} NMR (100.56 MHz, $C_6D_6$): $\delta = 135.3$ (Ar-CH), 135.2 (Ar-CH), 128.2 (Ar-CH), 128.1 (Ar-CH), 126.2 (Ar-CH), 126.1 (Ar-CH), 123.8 (CH), 123.2 (CH), 121.5 (CH), 120.5 (CH), 82.0 (C(CH$_3$)$_2$), 35.2 (N-CH$_3$), 34.1 (N-CH$_3$), 25.4 (CH$_3$) and 25.3 (CH$_3$); and $^{11}$B{$^1$H} NMR (128.3 MHz, $C_6D_6$): $\delta = 22.7$ (s), –10.1 (s) and –11.1 (s). HRMS (ESI): $m/z$ calculated for $C_{26}H_{34}B_3N_4O_2$: 467.2961 $[(M+H)]^+$; found: 467.2937.

**Synthesis of 5a.** Diphenylsilane (0.110 ml, 0.59 mmol) was added to a benzene (3.0 ml) solution of **2** (200 mg, 0.59 mmol), and the reaction mixture stirred for 1 h at room temperature. After removal of the solvent, the residue was washed with hexane and dried under vacuum to afford **5a** as a white powder (269 mg, 87%). Melting point: 182 °C (dec). $^1$H NMR (400 MHz, $C_6D_6$): $\delta = 7.95$ (d, 2H, $J = 7.2$ Hz, Ar-$H$), 7.66 (m, 2H, Ar-$H$), 7.62 (d, 2H, $J = 7.2$ Hz, Ar-$H$), 7.55–7.53 (m, 2H, Ar-$H$), 7.38–7.30 (m, 4H, Ar-$H$), 7.24–7.16 (m, 2H, Ar-$H$), $\delta = 7.14$–7.11 (m, 6H, Ar-$H$), 6.65 (s, 1H, C$H$), 6.56 (s, 1H, C$H$), 5.67 (s, 1H, C$H$), 5.59 (s, 1H, C$H$), 5.57 (s, 1H, Si$H$), 2.64 (s, 1H, N-C$H_3$) and 2.56 (s, 1H, N-C$H_3$); $^{13}$C{$^1$H} NMR (100.56 MHz, $C_6D_6$): $\delta = 139.3$ (Si$^qC$), 139.0 (Si$^qC$), 136.3 (Ar-CH), 136.0 (Ar-CH), 135.6 Ar-CH), 135.1 (Ar-CH), 128.4 (Ar-CH), 128.31 (Ar-CH), 128.25 (Ar-CH), 128.22 (Ar-CH), 127.9 (Ar-CH), 127.8 (Ar-CH), 126.6 (Ar-CH), 126.4 (Ar-CH), 123.5 (CH), 123.1 (CH), 121.8 (CH), 120.9 (CH), 35.2(N-CH$_3$) and 34.2 (N-CH$_3$); $^{11}$B{$^1$H} NMR (128.3 MHz, $C_6D_6$): $\delta = -9.2$ (br, two signals are overlapped). HRMS (ESI): $m/z$ calculated for $C_{32}H_{33}B_2N_4Si$: 523.2661 $[(M+H)]^+$; found: 523.2671.

**Synthesis of 5b.** Phenylsilane (0.091 ml, 0.74 mmol) was added to a benzene (3.0 ml) solution of **2** (250 mg, 0.74 mmol), and the reaction mixture stirred for 1 h at room temperature. After removal of the solvent, the residue was washed with hexane and dried under vacuum to afford **5b** as a yellow powder (280 mg, 85%). Melting point: 163 °C (dec). $^1$H NMR (400 MHz, $C_6D_6$): $\delta = 7.82$ (d, 2H, $J = 6.8$ Hz, Ar-$H$), 7.60 (d, 2H, $J = 6.8$ Hz, Ar-$H$), 7.35–7.29 (m, 6H, Ar-$H$), 7.23–7.19 (m, 2H, Ar-$H$), 7.11–7.09 (m, 3H, Ar-$H$), 6.71 (d, 1H, $J = 1.7$ Hz, C$H$), 6.63 (d, 1H, $J = 1.7$ Hz, C$H$), 5.75 (d, 1H, $J = 1.7$ Hz, C$H$), 5.73 (s, 1H, $J = 1.7$ Hz, C$H$), 4.82 (s, 1H, Si$H$), 4.77 (s, 1H, Si$H$), 2.75 (s, 3H, N-C$H_3$) and 2.63 (s, 3H, N-C$H_3$); $^{13}$C{$^1$H} NMR (100.56 MHz, $C_6D_6$): $\delta = 136.2$ (Si$^qC$), 136.0 (Ar-CH), 135.2 (Ar-CH), 135.0 (Ar-CH), 128.5 (Ar-CH), 128.3 (Ar-CH), 128.2 (Ar-CH), 127.8 (Ar-CH), 126.7 (Ar-CH), 126.3 (Ar-CH), 123.5 (CH), 122.6 (CH), 121.7 (CH), 121.4 (CH), 35.0 (N-CH$_3$) and 34.2 (N-CH$_3$); and $^{11}$B{$^1$H} NMR (128.3 MHz, $C_6D_6$): $\delta = -9.7$ (br, two signals are overlapped). HRMS (ESI): $m/z$ calculated for $C_{26}H_{29}B_2N_4Si$: 447.2348 $[(M+H)]^+$; found: 447.2368.

**Synthesis of 6.** Bis(4-fluorophenyl)phosphine (164 mg, 0.74 mmol) was added to a benzene (3.0 ml) and THF (0.5 ml) solution of **2** (250 mg, 0.74 mmol), and the reaction mixture stirred for 10 min at room temperature. After removal of the solvent, the residue was washed with hexane and dried under vacuum to afford **6** as a white powder (353 mg, 85%). Melting point: 210 °C. $^1$H NMR (400 MHz, $C_6D_6$): $\delta = 7.83$ (d, 2H, $J = 7.7$ Hz, Ar-$H$), 7.47–7.43 (m, 2H, Ar-$H$), 7.35 (t, 2H, $J = 7.3$ Hz Ar-$H$), 7.29–7.23 (m, 3H, Ar-$H$), 7.19–7.09 (m, 3H, Ar-$H$), 7.04 (d, 2H, $J = 7.3$ Hz, Ar-$H$), 6.76 (d, 1H, $J = 1.5$ Hz, C$H$), 6.67 (dd, 4H, $J = 14.9$, 8.2 Hz, Ar-$H$), 6.60 (d, 1H, $J = 1.5$ Hz, C$H$), 5.72 (d, 1H, $J = 1.5$ Hz, C$H$), 5.69 (s, 1H, $J = 1.5$ Hz, C$H$), 3.25 (s, 3H, N-C$H_3$) and 2.63 (s, 3H, N-C$H_3$); $^{13}$C{$^1$H} NMR (100.56 MHz, $C_6D_6$): $\delta = 136.6$ (d, $J = 7.2$ Hz, Ar-CH), 136.4 (d, $J = 7.2$ Hz, Ar-CH), 135.5 (Ar-CH), 134.8 (Ar-CH), 134.6 (Ar-CH), 128.1 (Ar-CH), 126.8 (Ar-CH), 126.6 (Ar-CH), 123.5 (CH), 123.1 (CH), 122.4 (CH), 120.8 (CH), 115.5 (d, $J = 6.3$ Hz, Ar-CH), 115.3 (d, $J = 6.3$ Hz, Ar-CH), 36.3 (d, $J = 18.6$ Hz, N-CH$_3$) and 34.7 (N-CH$_3$); $^{11}$B{$^1$H} NMR (128.3 MHz, $C_6D_6$): $\delta = -5.1$ (s) and –10.2 (s); $^{19}$F{$^1$H} NMR (376 MHz, $C_6D_6$): $\delta = -114.95$ (d, $J = 3.8$ Hz) and –115.98 (d, $J = 4.9$ Hz); and $^{31}$P{$^1$H} NMR (162 MHz, $C_6D_6$): $\delta = -34.1$. HRMS (ESI): $m/z$ calculated for $C_{32}H_{30}B_2N_4F_2P$: 561.2362 $[(M+H)]^+$; found: 561.2370.

**Synthesis of 7.** A benzene (2.0 ml) solution of **2** (200 mg, 0.59 mmol) was degassed using a freeze–pump–thaw method, and then $CO_2$ (1 bar) was introduced into the schlenk tube. After stirring for 5 min at room temperature, a white precipitate was collected by filtration and dried under vacuum to afford **7** as a white solid (172 mg, 76%). Melting point: 223 °C (dec). $^1$H NMR (400 MHz, CDCl$_3$): $\delta = 8.23$ (d, 2H, $J = 6.9$ Hz, Ar-$H$), 7.90 (d, 2H, $J = 6.9$ Hz, Ar-$H$), 7.47–7.35 (m, 6H, Ar-$H$), 6.99 (d, 1H, $J = 0.8$ Hz, C$H$), 6.95 (d, 1H, $J = 0.8$ Hz, C$H$), 6.57 (d, 2H, $J = 0.8$ Hz, C$H$), 3.27 (s, 3H, N-C$H_3$) and 3.13 (s, 3H, N-C$H_3$); $^{13}$C{$^1$H}

NMR (100.56 MHz, CDCl$_3$): $\delta = 135.1$ (Ar-CH), 133.1 (Ar-CH), 128.1 (Ar-CH), 128.0 (Ar-CH), 127.8 (Ar-CH), 127.3 (Ar-CH), 121.9 (CH), 121.4 (CH), 121.2 (CH), 120.5 (CH), 35.51 (N-CH$_3$) and 35.45 (N-CH$_3$); and $^{11}$B{$^1$H} NMR (128.3 MHz, CDCl$_3$): $\delta = -1.0$ (s) and $-10.8$ (s). Infrared $\nu$ cm$^{-1}$ (solid): 1667 (s). HRMS (ESI): $m/z$ calculated for $C_{21}H_{21}B_2N_4O$: 383.1851 $[(M+H)]^+$; found: 383.1858.

**Synthesis of 7-$^{13}$C.** By following the same procedure utilized for the synthesis of **7**, the reaction employing $^{13}CO_2$ afforded **7-$^{13}$C** (161 mg, 71%). Melting point: 223 °C (dec). $^{13}$C{$^1$H} NMR (100.56 MHz, CDCl$_3$): $\delta = 195.6$ ($C = O$), 135.0 (Ar-CH), 133.0 (Ar-CH), 128.0 (Ar-CH), 127.9 (Ar-CH), 127.7 (Ar-CH), 127.2 (Ar-CH), 121.8 (CH), 121.3 (CH), 121.2 (CH), 120.6 (CH), 35.43 (N-CH$_3$) and 35.36 (N-CH$_3$); and $^{11}$B{$^1$H} NMR (128.3 MHz, CDCl$_3$): $\delta = -0.8$ (s) and $-10.8$ (br). HRMS (ESI): $m/z$ calculated for $C_{20}{}^{13}CH_{21}B_2N_4O$: 384.1884 $[(M+H)]^+$; found: 384.1903.

**Synthesis of 8.** Styrene (0.085 ml, 0.74 mmol) was added to a benzene (3.0 ml) solution of **2** (250 mg, 0.74 mmol), and the reaction mixture stirred for 2 min at room temperature. After removal of the solvent, the residue was washed with hexane and dried under vacuum to afford **8** as a yellow powder (271 mg, 83%). Melting point: 238 °C. $^1$H NMR (400 MHz, $C_6D_6$): $\delta = 7.83$–7.82 (m, 4H, Ar-$H$), 7.44 (t, 2H, $J = 7.3$ Hz, Ar-$H$), 7.38–7.32 (m, 3H, Ar-$H$), 7.26 (t, 1H, $J = 7.3$ Hz, Ar-$H$), 7.13 (d, 2H, $J = 7.3$ Hz, Ar-$H$), 7.02 (t, 1H, $J = 7.3$ Hz, Ar-$H$), 6.84 (d, 1H, $J = 1.5$ Hz, C$H$), 6.56 (d, 2H, $J = 7.3$ Hz, Ar-$H$), 6.36 (d, 1H, $J = 1.5$ Hz, C$H$), 5.88 (d, 1H, $J = 1.5$ Hz, C$H$), 5.67 (s, 1H, $J = 1.5$ Hz, C$H$), 2.73 (dd, 1H, $J = 10.1$, 4.4 Hz, C$H$), 2.61 (s, 3H, N-C$H_3$), 2.55 (s, 3H, N-C$H_3$), 1.93 (dd, 1H, $J = 13.5$, 10.1 Hz, C$H$) and 1.04 (dd, 1H, $J = 13.5$, 4.4 Hz, C$H$); $^{13}$C{$^1$H} NMR (100.56 MHz, $C_6D_6$): $\delta = 155.6$ (Ar-$^qC$), 136.4 (Ar-CH), 135.2 (Ar-CH), 128.4 (Ar-CH), 128.1 (Ar-CH), 128.0 (Ar-CH), 127.8 (Ar-CH), 126.8 (Ar-CH), 126.4 (Ar-CH), 123.2 (CH), 121.1 (CH), 120.9 (CH), 120.1 (CH), 119.3 (CH), 35.3 (N-CH$_3$) and 34.7 (N-CH$_3$); and $^{11}$B{$^1$H} NMR (128.3 MHz, $C_6D_6$): $\delta = -5.8$ (s) and $-6.7$ (s). HRMS (ESI): $m/z$ calculated for $C_{28}H_{29}B_2N_4$: 443.2578 $[(M+H)]^+$; found: 443.2581.

**Synthesis of 9.** 1,1-Diphenylpropa-1,2-diene (114 mg, 0.59 mmol) was added to a benzene (2.0 ml) solution of **2** (200 mg, 0.59 mmol), and the reaction mixture stirred for 1 h at room temperature. After removal of the solvent, the residue was washed with hexane and dried under vacuum to afford **9** as a white powder (270 mg, 86%). Melting point: 245 °C. $^1$H NMR (400 MHz, $C_6D_6$): $\delta = 8.12$ (d, 2H, $J = 6.8$ Hz, Ar-$H$), 7.56 (d, 4H, $J = 7.4$ Hz, Ar-$H$), 7.46 (t, 2H, $J = 7.4$ Hz, Ar-$H$), 7.36 (t, 1H, $J = 7.4$ Hz, Ar-$H$), 7.21–7.17 (m, 5H, Ar-$H$), 7.12 (t, 2H, $J = 7.4$ Hz, Ar-$H$), 7.06–6.99 (m, 3H, Ar-$H$), 6.92 (t, 1H, $J = 7.4$ Hz, Ar-$H$), 6.50 (d, 1H, $J = 1.5$ Hz, C$H$), 6.26 (d, 1H, $J = 1.5$ Hz, C$H$), 6.16 (d, 1H, $J = 3.2$ Hz, C$H_2$), 5.68 (d, 1H, $J = 1.5$ Hz, C$H$), 5.45 (d, 1H, $J = 1.5$ Hz, C$H$), 5.14 (d, 1H, $J = 3.2$ Hz, C$H_2$), 2.70 (s, 3H, N-C$H_3$) and 2.43 (s, 3H, N-C$H_3$); $^{13}$C{$^1$H} NMR (100.56 MHz, $C_6D_6$): $\delta = 153.6$ (Ar-$^qC$), 152.0 (Ar-$^qC$), 136.6 (Ar-CH), 130.9 (Ar-CH), 130.2 (Ar-CH), 128.2 (Ar-CH), 127.4 (Ar-CH), 127.3 (Ar-CH), 126.9 (Ar-CH), 126.6 (Ar-CH), 124.1 (Ar-CH), 123.9 (CH), 122.8 (CH), 121.1 (CH), 119.8 (CH$_2$), 119.7 (CH) and 35.2 (overlap, N-CH$_3$); $^{11}$B{$^1$H} NMR (128.3 MHz, $C_6D_6$): $\delta = -3.5$ (s) and $-6.2$ (s). HRMS (ESI): $m/z$ calculated for $C_{35}H_{33}B_2N_4$: 531.2891 $[(M+H)]^+$; found: 531.2896.

**Synthesis of 10.** 1-Bromo-4-ethynylbenzene (80 mg, 0.44 mmol) was added to a benzene (2.0 ml) solution of **2** (150 mg, 0.44 mmol), and the reaction mixture was stirred for 1 h at room temperature. After removal of the solvent, the residue was washed with hexane and dried under vacuum to afford **10** as a yellow solid (183 mg, 80%). Melting point: 238 °C (dec). $^1$H NMR (400 MHz, $C_6D_6$): $\delta = 8.02$ (d, 2H, $J = 7.1$ Hz, Ar-$H$), 7.86 (s, 1H, C $=$ C$H$), 7.67 (d, 2H, $J = 7.1$ Hz, Ar-$H$), 7.48 (t, 2H, $J = 7.4$ Hz, Ar-$H$), 7.38 (t, 1H, $J = 7.4$ Hz, Ar-$H$), 7.34 (d, 2H, $J = 8.3$ Hz, Ar-$H$), 7.28–7.23(m, 3H, Ar-$H$), 7.13 (d, 2H, $J = 8.3$ Hz, Ar-$H$), 6.83 (s, 1H, C$H$), 6.68 (s, 1H, C$H$), 5.54 (s, 1H, C$H$), 5.50 (s, 1H, C$H$), 2.53 (s, 3H, N-C$H_3$) and 2.49 (s, 3H, N-C$H_3$); $^{13}$C{$^1$H} NMR (100.56 MHz, $C_6D_6$): $\delta = 149.1$ (Ar-$^qC$), 136.7 (Ar-CH), 135.3 (Ar-CH), 130.7 (Ar-CH), 129.4 (Ar-CH), 128.4 (Ar-CH), 127.9 (Ar-CH), 127.1 (Ar-CH), 126.7 (Ar-CH), 121.5 (CH), 121.3 (CH), 119.2 (CH), 118.7 (CH), 118.4 (Ar-$^qC$), 34.8 (N-CH$_3$) and 34.7 (N-CH$_3$); and $^{11}$B{$^1$H} NMR (128.3 MHz, CDCl$_3$): $\delta = -6.4$ (br, two signals are overlapped). HRMS (ESI): $m/z$ calculated for $C_{28}CH_{26}B_2BrN_4$: 519.1527 $[(M+H)]^+$; found: 519.1549.

**Synthesis of 11a.** Benzonitrile (0.061 ml, 0.59 mmol) was added into a benzene (2.0 ml) solution of **2** (200 mg, 0.59 mmol), and the reaction mixture was stirred for 5 min at room temperature. A white precipitate was collected by filtration and dried under vacuum to afford **11a** as a white solid (191 mg, 73%). Melting point: 156 °C (dec). $^1$H NMR (400 MHz, CDCl$_3$): $\delta = 8.33$ (d, 2H, $J = 7.3$ Hz, Ar-$H$), 7.61 (m, 2H, Ar-$H$), 7.50 (t, 2H, $J = 7.3$ Hz, Ar-$H$), 7.35–7.22 (m, 9H, Ar-$H$), 6.97 (s, 1H, C$H$), 6.95 (s, 1H, C$H$), 6.43 (s, 2H, C$H$), 3.27 (s, 3H, N-C$H_3$) and 3.08 (s, 3H, N-C$H_3$); $^{13}$C{$^1$H} NMR (100.56 MHz, CDCl$_3$): $\delta = 136.1$ (Ar-CH), 135.2 (Ar-CH), 128.5 (Ar-$^qC$), 127.8 (Ar-CH), 127.6 (Ar-CH), 127.5 (Ar-CH), 127.4 (Ar-CH), 127.2 (Ar-CH), 127.0 (Ar-CH), 126.5 (Ar-CH), 121.2 (CH), 120.9 (CH), 120.0 (CH),

119.6 (CH), 35.6 (N-CH$_3$) and 35.2 (N-CH$_3$); and $^{11}$B{$^1$H} NMR (128.3 MHz, CDCl$_3$): $\delta = -3.4$ (s) and $-8.2$ (s). HRMS (ESI): $m/z$ calculated for C$_{27}$H$_{26}$B$_2$N$_5$:442.2374 [$(M+H)$]$^+$; found: 442.2369.

**Synthesis of 11b.** 4-Chlorobenzonitrile (81 mg, 0.59 mmol) was added into a benzene (2.0 ml) solution of **2** (200 mg, 0.59 mmol), and the reaction mixture was stirred for 30 min at room temperature. A white precipitate was collected by filtration and dried under vacuum to afford **11b** as a white solid (197 mg, 70%). Melting point: 160 °C (dec). $^1$H NMR (400 MHz, CDCl$_3$): $\delta = 8.31$ (d, 2H, $J = 6.7$ Hz, Ar-H), 7.61 (m, 2H, Ar-H), 7.51 (t, 2H, $J = 7.5$ Hz, Ar-H), 7.33–7.29 (m, 6H, Ar-H), 7.19 (m, 2H, Ar-H), 7.00 (d, 1H, $J = 1.6$ Hz, CH), 6.98 (d, 1H, $J = 1.6$ Hz, CH), 6.44 (dd, 2H, $J = 2.6$ Hz and 1.7 Hz, CH), 3.27 (s, 3H, N-CH$_3$) and 3.05 (s, 3H, N-CH$_3$); $^{13}$C{$^1$H} NMR (100.56 MHz, CDCl$_3$): $\delta = 135.9$ (Ar-CH), 135.1 (Ar-CH), 129.1 (Ar-CH), 128.5 (Ar-$^q$C), 127.9 (Ar-CH), 127.8 (Ar-CH), 127.5 (Ar-CH), 127.1(Ar-CH), 126.7 (Ar-CH), 121.2 (CH), 120.9 (CH), 120.2 (CH), 119.7 (CH), 35.6 (N-CH$_3$) and 35.2 (N-CH$_3$); and $^{11}$B{$^1$H} NMR (128.3 MHz, CDCl$_3$): $\delta = -3.3$ (s) and $-8.5$ (s). HRMS (ESI): $m/z$ calculated for C$_{27}$H$_{25}$B$_2$ClN$_5$:476.1985 [$(M+H)$]$^+$; found: 476.1984.

**Data availability.** The data that support the findings of this study are available from the authors upon request.

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

## Acknowledgements

We gratefully acknowledge financial support from Nanyang Technological University and Singapore Ministry of Education (MOE2013-T2-1-005).

## Author contributions

B.W. performed the synthetic experiments Y.L. and R.G. performed the X-ray crystallographic measurements. H.H. conducted theoretical studies. R.K. conceived and supervised the study, and drafted the manuscript. All authors contributed to discussions.

## Additional information

**Accession codes.** CCDC 1442910–1442920 contain the supplementary crystallographic data for this paper. These data can be obtained free of charge from The Cambridge Crystallographic Data Centre via www.ccdc.cam.ac.uk/data_request/cif.

**Competing financial interests:** The authors declare no competing financial interests.

