## [Peer review file · Nature Communications]

Reviewers' Comments:

Reviewer #1 (Remarks to the Author)

- A. The work describes a series of reactions of cyclic bis-boron species with small molecules.
- B. The work is both highly interesting and novel.
- C. the work appears to be very thoroughly done. The figure of the structures is so small as to be illegible.
- D. N/A
- E. The discussion is too short.
- F. Is there computational evidence for the deformation of the FLP-type resonance form to a boat configuration?
- G. The authors do not reference a related "aromatic" system that reacts with H₂ (JACS 2014, 36, 13453).
- H. I find the discussion ends rather abruptly without sufficient consideration.

Reviewer #2 (Remarks to the Author)

In this manuscript, Kinjo et al. reported the synthesis of 1,4,2,5-diazadiborinine derivative, in which the two boron atoms exhibit the Lewis acidic and basic properties, respectively. Although the two boron atoms in this molecule are spectroscopically equivalent confirmed by both experimental and computational studies, the further reactivity studies show that the molecule can undergo heterolytic cleavages of C-O, B-H, Si-H, and P-H bonds as well as [4 + 2] cycloaddition reaction with the C=O, C=C, C≡C, and C≡N bonds, which indicates the ambiphilic nature of 1,4,2,5-diazadiborinine derivative. In addition, the structure of 1,4,2,5-diazadiborinine derivative is remarkably different from those reported in their previous papers (Chem. Sci., 2015, 6, 2893-2902; Angew. Chem. Int. Ed. 2014, 53, 9280-9283; Science, 2011, 333, 610-613; Nat. Commun., 2015, 6, 7340; Chem. Sci., 2015, 6, 7150-7155), and the results are reasonable and interesting, so I recommend its publication after minor revision.

Major concerns:

1. For compound 3, the author stated that the compound 3 was obtained as a mixture of diastereomers (1:1), but the calculated results provided in Supporting Information indicated that the energy gap of 3A and 3B is 1.5 kcal/mol, which should be too large. Maybe the authors use the Gibbs free energy, which is more reasonable in this case.
2. For compounds 4 and 5, the author stated that the compound 4 (and 5) was obtained as the solo diastereomer, but the computational outcomes show that the energy gaps of 4 and 5 are both too narrow. In addition, they did not provide the Gibbs free energy of 4-A, which can be concluded to be -1454.745443 a.u. In this case, the authors should point out that it should not be the thermodynamically controlled reaction.
3. The author should explain why the (FC₄H₆)₂P group and the H atom point on the B atom in 6 are different from those of in 3~5, and some reasons for understanding the origin of the stereoselectivity should be given.

4. In the [4 + 2] cycloaddition processes, the authors should also perform some simple calculations to explore the regioselectivity and stereoselectivity.

Minor concerns:

1. Some references should be listed all the authors, such as 8, 16, 17, 19, 28, 34, 36, 38.
2. Reference 17 should be updated.

Reviewer #3 (Remarks to the Author)

A. Summary of the key results

Kinjo and coworkers present the synthesis of an electron rich B₂N₂C₂ heterocycle and its reactivity with small molecules, leading to bond activation and cycloadditions.

B. Originality and interest: if not novel, please give references

Here it needs to be pointed out that the authors themselves have reported a very similar compound in two publications, references 35 and 36 in the text. The first of these was in Nature Communications itself. The new heterocycle is slightly different from the previous one, most obviously in the connectivity of the atoms (NBCNBC vs. NBNBCB). Thus, the new compound is much more symmetrical where the previous compound showed distinctly inequivalent boron atoms, one electron rich, one electron poor. The fact that the new compound also does similar chemistry raises interesting questions about whether the borons really are "+/-". The new compound is also shown to activate some slightly more challenging bonds than the previous system.

However, while the new results are definitely interesting, the unmistakable similarity of the systems cannot be ignored, and the concepts involved are nearly identical. The manuscript definitely does not add enough novelty to be publishable in a journal of this quality. This can also be seen in the way the manuscript is written, for instance, the first mention of the new chemistry begins with "Extending this strategy, herein...". To me this is a clear case of a manuscript unsuited to Nature Communications, although it would be well at home in a specialized organic or inorganic chemistry journal.

C. Data & methodology: validity of approach, quality of data, quality of presentation

The work appears to be technically sound.

D. Appropriate use of statistics and treatment of uncertainties

No problems here.

E. Conclusions: robustness, validity, reliability

Appear to be valid.

F. Suggested improvements: experiments, data for possible revision

I cannot think of any improvements that would make the manuscript publishable in this journal.

G. References: appropriate credit to previous work?

References are appropriate.

H. Clarity and context: lucidity of abstract/summary, appropriateness of abstract, introduction and conclusions

Everything fine here, but inherently lacks the required novelty.

Reviewer #4 (Remarks to the Author)

This manuscript describes the synthesis of a novel fused boron-containing heterocyclic molecule, 1,4,2,5-diazaborinine. The authors have demonstrated that the title compound exhibits characteristics of aromatic compounds (i.e., planarity, bond homologation, and ring current as suggested by NICS values) but it is nevertheless quite reactive toward small molecules. For example, the authors nicely demonstrated that the title molecule has chemically equivalent, ambiphilic boron atoms which act as both a nucleophile and an electrophile to small molecules with activatable bonds such as silanes, alkynes, nitriles, and interestingly, boranes. The reaction products are well characterized, including X-ray structures. The experimentally observed reactivity is consistent with the electronic structure determination using DFT methods. The authors characterize the observed reactivity in the vein of frustrated Lewis pair (FLP) chemistry.

This is nice work, and I therefore recommend publication in Nature Communications, however only after revisions.

1) A similar C₂B₂N₂ heterocycle is known, which should be cited: JACS 2009, 131, 5858-65.

2) The authors state: "The resonance stabilization energy (RSE) value of 2' is approximately 12.2 kcal/mol greater than that of benzene (34.1 kcal/mol)".

This statement is very misleading as a direct comparison of the fused polycyclic title compound with the monocyclic benzene cannot be used by the authors' employed method to evaluate the RSE of the of the six-membered BN heterocyclic core. The origin of the authors' results likely lie in the extended conjugated nature of the title compound 2' which will be additionally destabilized upon hydrogenation.

A more appropriate determination of RSE of the key B₂N₂C₂ heterocyclic core would be to use the unsubstituted compound (forth compound in Figure 2).

3) The authors state: "In the solid state structures of 4 and 5a, Bpin or Ph₂HSi group and the H atom on the B atom are attached on the same side of the six-membered B₂C₂N₂ ring. By contrast, the (FH₄C₆)₂P group and the H atom on the B atom in 6 point in opposite direction".

Rather than just stating the experimental observation, the manuscript would be significantly improved if the authors can describe the differences in the nature of the mechanism of the X-H activation and support their hypothesis with data (e.g., calculations).

4) The high diastereoselectivity for the addition of styrene to the title compound is interesting because it is counterintuitive. It appears that the phenyl group is pointing toward the more sterically encumbered direction (clashing with N-Me and avoiding the smaller C-H). An explanation plus support would be appropriate.

Other typos:

- instead of "solo diastereomer" I recommend "single diastereomer"
- The "Discussion" section is only one sentence. It maybe more appropriate to replace it with "Conclusion".

To *Nature Communication*

21-Apr-2016

Response to referees comments

We thank all referees and editors for the feedback and insightful comments, and have incorporated all of their suggestions into the manuscript, which are outlined below point-by-point and highlighted in yellow in the revised manuscript. Specifically:

Reviewer #1:

(1)

A. The work describes a series of reactions of cyclic bis-boron species with small molecules.

B. The work is both highly interesting and novel.

We thank the reviewer for concisely pointing out the key results and rating as both highly interesting and novel.

(2)

C. the work appears to be very thoroughly done. The figure of the structures is so small as to be illegible.

We have increased the size of the Figures 1b, 1d and 3b.

(3)

D. N/A

E. The discussion is too short.

We have expanded the discussion section, in which the relationship to previous publications in the field of small molecule activation by aromatic compounds is discussed, with highlighting similarities and differences. In addition, concluding remark and perspective are included in the section.

(4)

F. *Is there computational evidence for the deformation of the FLP-type resonance form to a boat configuration?*

We have performed DFT calculation regarding the reaction mechanism for the formation of compounds **4**, **5a**, **8** and **9**. At those transition states, distortion of the B₂C₂N₂ six-membered ring moiety of **2** is observed (Figure 4) and it displays slight boat configuration character. The mild deformation is probably due to the lack of flexibility of the B₂C₂N₂ ring skeleton. We guess that this distortion may enhance zwitterion nature at the two boron centers of **2**. These results are added in the sections of “Proposed mechanism based on DFT calculation” and “Discussion” in main text.

(5)

G. *The authors do not reference a related "aromatic" system that reacts with H₂ (JACS 2014, 36, 13453).*

We have added the reference (JACS **2014**, *136*, 13453) in ref 54.

We also added a reference related to [4+2] cycloaddition between a boron-containing 6-membered ring and alkyne (JACS **2015**, *137*, 13264) in ref 56.

(6)

H. *I find the discussion ends rather abruptly without sufficient consideration.*

Related to the response to “E”, we have expanded the discussion section in which key results, comparison to other reports, and concluding remarks are involved.

Reviewer #2:

In this manuscript, Kinjo et al. reported the synthesis of 1,4,2,5-diazadiborinine derivative, in which the two boron atoms exhibit the Lewis acidic and basic properties, respectively. Although the two boron atoms in this molecule are spectroscopically equivalent confirmed by both experimental and computational studies, the further reactivity studies show that the molecule can undergo heterolytic cleavages of C-O, B-H, Si-H, and P-H bonds as well as [4 + 2] cycloaddition reaction with the C=O, C=C, C≡C, and C≡N bonds, which indicates the ambiphilic nature of 1,4,2,5-diazadiborinine derivative. In addition, the structure of 1,4,2,5-diazadiborinine derivative is remarkably different from those reported in their previous papers (Chem. Sci., 2015, 6, 2893-2902; Angew. Chem. Int. Ed. 2014, 53, 9280-9283; Science, 2011, 333, 610-613; Nat. Commun., 2015, 6,

Assistant Professor Rei KINJO

Division of Chemistry and Biological Chemistry (06-20)

School of Physical and Mathematical Sciences

Nanyang Technological University

21 Nanyang Link Singapore 637371

Tel: (65)6592-2625 (GMT+8h)

Email: RKinjo@ntu.edu.sg

7340; *Chem. Sci.*, 2015, 6, 7150-7155), and the results are reasonable and interesting, so I recommend its publication after minor revision.

We are grateful to the reviewer for his/her careful reading and the high number of critical comments.

Major concerns:

(1)

For compound 3, the author stated that the compound 3 was obtained as a mixture of diastereomers (1:1), but the calculated results provided in Supporting Information indicated that the energy gap of 3A and 3B is 1.5 kcal/mol, which should be too large. Maybe the authors use the Gibbs free energy, which is more reasonable in this case.

We re-calculated the energy gap of **3A** and **3B**, using the Gibbs free energy, in which dispersion force is also considered. The result indicates that the energy gap between **3A** and **3B** is 0.2 kcal/mol. We updated the result in the Supporting Information (Supplementary Figure 57), and it is also shown in Fig 5a.

(2)

For compounds 4 and 5, the author stated that the compound 4 (and 5) was obtained as the solo diastereomer, but the computational outcomes show that the energy gaps of 4 and 5 are both too narrow. In addition, they did not provide the Gibbs free energy of 4-A, which can be concluded to be -1454.745443 a.u. In this case, the authors should point out that it should not be the thermodynamically controlled reaction.

We agree with the reviewer's opinion. For the formation of compounds **4** and **5**, the reactions should not be the thermodynamically controlled reaction. We added a sentence noting this point, in the section of "Proposed mechanism based on DFT calculation" in main text on page 8.

(3)

The author should explain why the (FC₄H₆)₂P group and the H atom point on the B atom in 6 are different from those of in 3~5, and some reasons for understanding the origin of the stereoselectivity should be given.

For the formation of compound **6**, we envisage that reaction proceeds in stepwise manner. Thus, the initial step is protonation of **2** via a heterolytic cleavage of the P-H bond of (FC₄H₆)₂P-H to afford

an ionic intermediate involving cationic protonated-**2** and $(\text{FC}_4\text{H}_6)_2\text{P}$ anion. Because **6** is less stable (0.9 kcal/mol) than the other isomer **6*** (Fig. 5b), this reaction should not be the thermodynamically controlled reaction, and thus **6** is kinetic product. Since we could confirm only a concerted pathway by DFT calculations (Supplementary Figure 59), actual reaction mechanism for the diastereoselective formation of **6** remains unclear. Experimental observation and isolation of ionic intermediate such as **INT-3** and **INT-6** in fig 5, is a subject of our ongoing study.

Meanwhile, for the formation of **4** and **5**, reactions seem to proceed in concerted manner as supported by DFT calculation (Figure 4a and 4b), which affords the products featuring both the H atom and E (E = Bpin or $\text{Ph}_n\text{SiH}_{(3-n)}$) group on the same side of the $\text{B}_2\text{C}_2\text{N}_2$ six-membered ring.

In H-Bpin and $\text{Ph}_n\text{SiH}_{(4-n)}$, the H atom is hydridic instead of protic. Therefore, it is inferred that an ionic intermediate involving anionic **2**-hydride adduct and Bpin cation (or $\text{Ph}_n\text{SiH}_{(3-n)}$ cation) $[\text{2-H}]^- [\text{Bpin (or } \text{Ph}_n\text{SiH}_{(3-n)})]^+$ would not be generated because of the instability of the boryl or silyl cation. Therefore, the reactions with H-Bpin and $\text{Ph}_n\text{SiH}_{(4-n)}$ would not be stepwise but concerted, which could be, at least in part, the origin of those stereo-selectivity.

We added a paragraph describing the proposed mechanism and stereo-selectivity, in the section of “Proposed mechanism based on DFT calculation” in main text.

(4)

In the [4 + 2] cycloaddition processes, the authors should also perform some simple calculations to explore the regioselectivity and stereoselectivity.

We carried out calculations to explore the reaction mechanism for the formation of **8** and **9**, and found the energetically reasonable reaction pathway for the region-selective formation of **8** and stereo-selective formation of **9**. We observed important role of dispersion interaction in determining the region selectivity (Supplementary Table 6). These results are included in Fig. 4, and described, in the section of “Proposed mechanism based on DFT calculation” in main text on page 8-9.

(5)

Minor concerns:

- 1. Some references should be listed all the authors, such as 8, 16, 17, 19, 28, 34, 36, 38.*
- 2. Reference 17 should be updated.*

We have listed all the authors in all references, and updated ref 17.

Reviewer #3:

(1)

A. Summary of the key results:

Kinjo and coworkers present the synthesis of an electron rich B₂N₂C₂ heterocycle and its reactivity with small molecules, leading to bond activation and cycloadditions.

B. Originality and interest: if not novel, please give references:

Here it needs to be pointed out that the authors themselves have reported a very similar compound in two publications, references 35 and 36 in the text. The first of these was in Nature Communications itself. The new heterocycle is slightly different from the previous one, most obviously in the connectivity of the atoms (NBCNBC vs. NBNCBC). Thus, the new compound is much more symmetrical where the previous compound showed distinctly inequivalent boron atoms, one electron rich, one electron poor. The fact that the new compound also does similar chemistry raises interesting questions about whether the borons really are "+/-". The new compound is also shown to activate some slightly more challenging bonds than the previous system. However, while the new results are definitely interesting, the unmistakable similarity of the systems cannot be ignored, and the concepts involved are nearly identical. The manuscript definitely does not add enough novelty to be publishable in a journal of this quality. This can also be seen in the way the manuscript is written, for instance, the first mention of the new chemistry begins with "Extending this strategy, herein...". To me this is a clear case of a manuscript unsuited to Nature Communications, although it would be well at home in a specialized organic or inorganic chemistry journal.

We acknowledge that the reviewer spent a substantial amount of time and effort to review our manuscript as well as previous our reports and gave valuable inputs to improve further the quality of our paper. We have omitted the sentence "Extending this strategy" in the revised manuscript.

(2)

C. Data & methodology: validity of approach, quality of data, quality of presentation:

The work appears to be technically sound.

Assistant Professor Rei KINJO

Division of Chemistry and Biological Chemistry (06-20)

School of Physical and Mathematical Sciences

Nanyang Technological University

21 Nanyang Link Singapore 637371

Tel: (65)6592-2625 (GMT+8h)

Email: RKinjo@ntu.edu.sg

D. Appropriate use of statistics and treatment of uncertainties:

No problems here.

E. Conclusions: robustness, validity, reliability:

Appear to be valid.

F. Suggested improvements: experiments, data for possible revision:

I cannot think of any improvements that would make the manuscript publishable in this journal.

G. References: appropriate credit to previous work?:

References are appropriate.

H. Clarity and context: lucidity of abstract/summary, appropriateness of abstract, introduction and conclusions:

Everything fine here, but inherently lacks the required novelty.

We acknowledge referee's very positive note about our data and methodology, statistics~, conclusion, references, and clarity and context.

Reviewer #4:

This manuscript describes the synthesis of a novel fused boron-containing heterocyclic molecule, 1,4,2,5-diazaborinine. The authors have demonstrated that the title compound exhibits characteristics of aromatic compounds (i.e., planarity, bond homologation, and ring current as suggested by NICS values) but it is nevertheless quite reactive toward small molecules. For example, the authors nicely demonstrated that the title molecule has chemically equivalent, ambiphilic boron atoms which act as both a nucleophile and an electrophile to small molecules with activatable bonds such as silanes, alkynes, nitriles, and interestingly, boranes. The reaction products are well characterized, including X-ray structures. The experimentally observed reactivity is consistent with the electronic structure determination using DFT methods. The authors characterize the observed reactivity in the vein of frustrated Lewis pair (FLP) chemistry.

This is nice work, and I therefore recommend publication in Nature Communications, however only after revisions.

We thank reviewer's comment highlighting the significance of present work.

(1)

Assistant Professor Rei KINJO
Division of Chemistry and Biological Chemistry (06-20)
School of Physical and Mathematical Sciences
Nanyang Technological University
21 Nanyang Link Singapore 637371
Tel: (65)6592-2625 (GMT+8h)
Email: RKinjo@ntu.edu.sg

A similar C₂B₂N₂ heterocycle is known, which should be cited: *JACS* 2009, 131, 5858-65.

We have cited the reference (*JACS* 2009, 131, 5858) in ref 58, and relevant references in ref 57 and 59.

(2)

The authors state: "The resonance stabilization energy (RSE) value of 2' is approximately 12.2 kcal/mol greater than that of benzene (34.1 kcal/mol)". This statement is very misleading as a direct comparison of the fused polycyclic title compound with the monocyclic benzene cannot be used by the authors' employed method to evaluate the RSE of the of the six-membered BN heterocyclic core. The origin of the authors' results likely lie in the extended conjugated nature of the title compound 2' which will be additionally destabilized upon hydrogenation. A more appropriate determination of RSE of the key B₂N₂C₂ heterocyclic core would be to use the unsubstituted compound (forth compound in Figure 2).

We have recalculated the RSE of the B₂N₂C₂ heterocyclic core using the un-substituted compound, thus parent 1,3,2,5-diazadiborinine, which revealed that the RSE value of parent 1,3,2,5-diazadiborinine is 37.9 kcal/mol smaller than that of benzene. This result has been updated in main text on page 5, and relevant references are added as ref 38 and 39. We also updated the Supporting Information (Supplementary Table 4).

(3)

The authors state: "In the solid state structures of 4 and 5a, Bpin or Ph₂HSi group and the H atom on the B atom are attached on the same side of the six-membered B₂C₂N₂ ring. By contrast, the (FH₄C₆)₂P group and the H atom on the B atom in 6 point in opposite direction". Rather than just stating the experimental observation, the manuscript would be significantly improved if the authors can describe the differences in the nature of the mechanism of the X-H activation and support their hypothesis with data (e.g., calculations).

Related to the response to the comment (3) by the reviewer #2, we have performed DFT calculation to explore the reaction pathways for the formation of compounds **4** and **5a**. Calculation results show that these reactions proceed through a concerted pathway (Fig. 4), which could be the reason for the stereo-selective formation of **4** and **5a**.

In contrast, we infer that compound **6** is formed by stepwise mechanism because the $(\text{FH}_4\text{C}_6)_2\text{P}$ group and the H atom on the B atom point in opposite direction. The initial step could be protonation of **2** concomitant with a heterolytic cleavage of the P-H bond of $(\text{FC}_4\text{H}_6)_2\text{P-H}$ to afford an ionic intermediate $[\text{2-H}]^+[(\text{FC}_4\text{H}_6)_2\text{P}]^-$. Then, intermediate $[\text{2-H}]^+[(\text{FC}_4\text{H}_6)_2\text{P}]^-$ would undergo a P-B bond formation. The second step should be the kinetically controlled reaction as experimentally obtained **6** is less stable (0.9 kcal/mol) than the other diastereomer **6***. However, because we could confirm only a concerted pathway by DFT calculations, actual reaction mechanism for the diastereoselective formation of **6** remains unclear at present. Accordingly, challenge to isolate the proposed ionic intermediate such as **INT-3** and **INT-6** in figure 5, will be a subject of our future study. We added a section describing this point in the section of “Proposed mechanism based on DFT calculation” in main text.

(4)

The high diastereoselectivity for the addition of styrene to the title compound is interesting because it is counterintuitive. It appears that the phenyl group is pointing toward the more sterically encumbered direction (clashing with N-Me and avoiding the smaller C-H). An explanation plus support would be appropriate.

We performed calculation to explore the reaction pathway for the diastereo-selective formation of **8**, and revealed that formation of **8** is favoured both thermodynamically and kinetically (Fig. 4c). Thus, the product **8** is more stable (0.6 kcal/mol) than its diastereomer **8***, and the activation barrier for the formation of **8** is 1.9 kcal/mol smaller than that of **8***.

As shown in fig. 1d, both the HOMO and LUMO of **2** have significant amplitude between the B-C moieties rather than the B-N moieties of the $\text{B}_2\text{C}_2\text{N}_2$ six-membered ring. In the reaction of **2** and styrene, hence, in order to maximize interaction between the frontier orbitals of **2** and π or π^* orbitals of styrene, the C=C bond of styrene does not lie completely parallel to the line connecting the two B atoms of **2**, but are slightly directed towards the midpoints of two B-C bonds of **2**. In such transition state, directing the phenyl group of styrene toward the B atom of **2** would cause significant steric repulsion between the Ph rings of styrene and **2**. The transition state for the favored

pathway looks less sterically encumbered, which could also contribute to the diastereoselectivity of the cycloaddition.

We describe this point in the sections of "Proposed mechanism based on DFT calculation" in main text on page 8.

(5)

Other typos:

- *instead of "solo diastereomer" I recommend "single diastereomer"*

We have corrected the sentence, on page 6.

- *The "Discussion" section is only one sentence. It maybe more appropriate to replace it with "Conclusion".*

We have expanded the discussion section, which includes the relationship to previous reports, with highlighting similarities and differences, concluding remark and perspective.

Editorial requests:

(1)

Alongside addressing the technical concerns raised by the reviewers we would also recommend that the discussion section of the manuscript is expanded, and that the relationship to your other publications in the field is clearly and transparently discussed, highlighting similarities and differences.

We have expanded the discussion section.

(2)

At the same time, we ask that you ensure your manuscript complies with our format requirements, which are summarized in the following checklist.

We checked the check list and followed the format.

School of Physical and Mathematical Sciences

Reg. No. 200604393R

Thank you very much for taking your valuable time for reviewing our manuscript.

Best regards

Rei Kinjo

Assistant Professor Rei KINJO
Division of Chemistry and Biological Chemistry (06-20)
School of Physical and Mathematical Sciences
Nanyang Technological University
21 Nanyang Link Singapore 637371
Tel: (65)6592-2625 (GMT+8h)
Email: RKinjo@ntu.edu.sg

Reviewers' Comments:

Reviewer #2 (Remarks to the Author)

I think that the authors have responded to all my questions, and the quality of the revised manuscript has been obviously improved, so I recommend its publication in Nature Communications as soon as possible.

Reviewer #4 (Remarks to the Author)

In this revised manuscript, the authors sufficiently addressed the concerns that I and other reviewers raised for the original submission. There is one clarification that I encourage the authors to address prior to acceptance:

The authors now state: "The resonance stabilization energy (RSE) value of parent 1,4,2,5-diazadiborinine 2 estimated at the B3LYP/6-311+G(d,p) level is 37.9 kcal/ mol smaller than that of benzene (Supplementary Table 4)" based on a revised calculation. In view that benzene's RSE is ~32-34 kcal/mol, the authors should address what the value "37.9 kcal/mol less than benzene's RSE" means in terms of 1,4,2,5-diazadiborinine's aromatic character and how this is or not consistent with the reactivity reported in the manuscript.

To *Nature Communication*

5-May-2016

Dear Dr. Luke Batchelor

Please find herewith our response to referee comments. They are indicated using “track change” in the revised manuscript. Specifically:

Reviewer #4:

*In this revised manuscript, the authors sufficiently addressed the concerns that I and other reviewers raised for the original submission. There is one clarification that I encourage the authors to address prior to acceptance: The authors now state: "The resonance stabilization energy (RSE) value of parent 1,4,2,5-diazadiborinine **2** estimated at the B3LYP/6-311+G(d,p) level is 37.9 kcal/mol smaller than that of benzene (Supplementary Table 4)" based on a revised calculation. In view that benzene's RSE is ~32-34 kcal/mol, the authors should address what the value "37.9 kcal/mol less than benzene's RSE" means in terms of 1,4,2,5-diazadiborinine's aromatic character and how this is or not consistent with the reactivity reported in the manuscript.*

We added the following sentence on the page 5:

“Since the RSE value of benzene is estimated to be about 34 kcal mol⁻¹, although the reported values vary depending on the method used,^{38,39} it can be inferred that the aromatic character of parent 1,4,2,5-diazadiborinine **2**” is significantly weak, in line with the less negative NICS values for **2**” than that of benzene (Fig. 2). “

We also added the following sentence in the conclusion paragraph (on page 11):

“According to the NICS values of **2**, **2'**, **2**” and benzene as well as the gap of the RSE values between **2**” and benzene, the annulation of the B₂C₂N₂ ring in 1,4,2,5-diazadiborinine may increase the aromatic nature of **2'**, which is comparable to that of benzene. Nevertheless, **2** exhibits reactivity that is peculiar and much higher than that of benzene.”

Best regards

Rei Kinjo

Assistant Professor Rei KINJO
Division of Chemistry and Biological Chemistry (06-20)
School of Physical and Mathematical Sciences
Nanyang Technological University
21 Nanyang Link Singapore 637371
Tel: (65)6592-2625 (GMT+8h)
Email: RKinjo@ntu.edu.sg